# Identification and Construction of Strong Promoters in *Yarrowia lipolytica* Suitable for Glycerol-Based Bioprocesses

**DOI:** 10.3390/microorganisms11051152

**Published:** 2023-04-28

**Authors:** Ioannis Georgiadis, Christina Tsiligkaki, Victoria Patavou, Maria Orfanidou, Antiopi Tsoureki, Aggeliki Andreadelli, Eleni Theodosiou, Antonios M. Makris

**Affiliations:** 1Institute of Applied Biosciences, Centre for Research & Technology Hellas (CERTH), 57001 Thessaloniki, Greece; georgiais@bio.auth.gr (I.G.); tsiligkakichri@certh.gr (C.T.); patavouvictoria@gmail.com (V.P.); mariaorfan97@certh.gr (M.O.); adatsoureki@certh.gr (A.T.); a.andreadelli@certh.gr (A.A.); eleni.theodosiou@certh.gr (E.T.); 2School of Biology, Faculty of Sciences, Aristotle University of Thessaloniki, 54636 Thessaloniki, Greece; 3Department of Chemical Engineering, School of Engineering, Aristotle University of Thessaloniki, 54636 Thessaloniki, Greece

**Keywords:** synthetic biology, promoter engineering, UAS, *Yarrowia lipolytica*, LIP2 lipase, glycerol, biofuel

## Abstract

*Yarrowia lipolytica* is a non-pathogenic aerobic yeast with numerous industrial biotechnology applications. The organism grows in a wide variety of media, industrial byproducts, and wastes. A need exists for molecular tools to improve heterologous protein expression and pathway reconstitution. In an effort to identify strong native promoters in glycerol-based media, six highly expressed genes were mined from public data, analyzed, and validated. The promoters from the three most highly expressed (H3, ACBP, and TMAL) were cloned upstream of the reporter mCherry in episomal and integrative vectors. Fluorescence was quantified by flow cytometry and promoter strength was benchmarked with known strong promoters (pFBA1in, pEXP1, and pTEF1in) in cells growing in glucose, glycerol, and synthetic glycerol media. The results show that pH3 > pTMAL > pACBP are very strong promoters, with pH3 exceeding all other tested promoters. Hybrid promoters were also constructed, linking the Upstream Activating Sequence 1B (UAS1B8) with H3(260) or TMAL(250) minimal promoters, and compared to the UAS1B8-TEF1(136) promoter. The new hybrid promoters exhibited far superior strength. The novel promoters were utilized to overexpress the lipase *LIP2*, achieving very high secretion levels. In conclusion, our research identified and characterized several strong *Y. lipolytica* promoters that expand the capacity to engineer *Yarrowia* strains and valorize industrial byproducts.

## 1. Introduction

The global consumption of biodiesel and biojet fuel is constantly increasing to meet the demands of the so-desired sustainable bioeconomy [1]. However, biodiesel production does not come without drawbacks. During the saponification and transesterification of oils and fats for its production, the major byproduct, crude glycerol, is produced at a ratio of 1:10 to biodiesel. As biodiesel production increases, the accumulating surplus of crude glycerol calls for value-added strategies for its efficient and sustainable disposal. Toward this goal, many biodiesel companies are trying to design and develop novel bioprocesses to valorize this valuable feedstock, ensuring in this way the economic viability of biodiesel [2,3,4,5]. One of the most potent microorganisms to utilize this renewable carbon source is *Yarrowia lipolytica*, a non-pathogenic (classified as GRAS) aerobic yeast that has been a biotechnological workhorse for the production of proteins and organic acids for more than 70 years [5,6,7,8,9,10,11,12,13]. Despite the progress that has led to cost-effective and competitive glycerol-based production processes, *Y. lipolytica* strains will still need, in many cases, further improvement to meet the requirements of an industrial commercial bioprocess [12,14]. Systems metabolic engineering, that combines tools and strategies from synthetic and systems biology as well as evolutionary engineering, has not only greatly improved the performance and robustness of strains with industrial relevance, but has also allowed them to be tailor-made by redesigning relevant biosynthetic pathways [15,16,17]. Such approaches expand the possibilities of industrial applications, which range from the production of tailor-made lipids and biofuels to whole-cell biocatalysts and biosensors [18,19,20,21].

Over the past two decades, numerous molecular tools to engineer *Y. lipolytica* have been developed, taking advantage of the availability of genome information. Transformation methodologies were established, episomal and integrative vectors constructed, and gene knockouts via recombination or CRISPR were performed [22,23,24]. One impediment, though, is that unlike *Saccharomyces cerevisiae*, where episomal vectors can be maintained stably and in numerous copies, in *Y. lipolytica* they are present only in few copies, making the genomic integration of expression elements necessary for useful engineered strains. In such integrated expression cassette systems, the utilization of strong promoters that will be active under the desirable process conditions is even more imperative. One such example of a strong promoter in *Y. lipolytica* is the XPR2 promoter, which encodes an inducible alkaline extracellular protease; however, its promoter strength is affected by numerous parameters, such as pH, type of carbon and nitrogen source, presence of peptones, etc. [25]. Although there is no single comprehensive report comparing all available cloned promoters, either using episomal or integrative vectors, the relative potency of several of them has recently been compared. Using flow cytometry to determine the expression level of the *hrGFP* reporter gene, the strength of promoters ranked as follows: EXP1 > TEF1 > GPD > GPAT > YAT1 > XPR2 > FBA1 [26,27]. Another research study ranked TEF1intron >> GPD > EXP1 > TEF1 > ILV5 > YAT1 > ICL > FBA1, GPAT, DGA1, GPM1, and FBA1intron = 0 [23]. Another strategy to enhance gene expression involves maintaining an intron with its corresponding promoter. This may be especially relevant in *Y. lipolytica* as 15% of its genes contain introns. It was reported that the pFBA1in promoter was five-fold stronger than the pFBA1 promoter [28] and that the pTEF1in promoter increased expression levels 17-fold over the pTEF1 promoter [29]. It is postulated that introns enhance gene expression because they contain regulatory elements, facilitate mRNA export, or increase the rate of transcription initiation [30].

To further increase promoter strength, a synthetic biology approach for promoter engineering was effectively pursued in *Y. lipolytica*. The identification of distal Upstream Activating Sequences (UAS), namely the 105 bp UAS1B of the XPR2 promoter, has led to the development of hybrid promoters by fusing copies of UAS to minimal promoters [25]. Independent of the core promoter element utilized, the hybrid promoters exhibit an increase in expression efficiency that is proportional to the number of fused UAS repeats [26,31]. Apart from the enhancer regions (UAS), it has been shown that core elements and TATA box sequences of endogenous *Y. lipolytica* promoters can be used in modularity to increase promoter strength and transcription levels [32].

In our effort to develop expression systems optimized for growth in glycerol-based media, we isolated and characterized novel promoter elements able to drive high gene expression. The three highest-ranking elements were cloned and fused to reporter genes in both episomal and integrative vectors. The expression cassettes were introduced in different *Y. lipolytica* strains and expression was quantified by flow cytometry. In parallel, expression was compared with the known strong promoters TEF1in, EXP1, and FBA1in [26,28]. All three novel promoters (pH3, pTMAL, and pACBP) were found to be exceptionally strong in both glycerol and glucose media in the strains tested, with H3 exceeding the best-performing known promoter. Hybrid promoters were also constructed, based on minimal promoter elements from the newly characterized promoters, in an effort to increase promoter strength. The hybrid UAS1B8-TMAL promoter was superior to the UAS1B8-TEF1 hybrid promoter. Finally, recombinant *Y. lipolytica* strains containing newly constructed vectors bearing the strongest promoters (native or hybrid) and the native *LIP2* gene (one or multiple copies) were evaluated for their ability to produce the extracellular LIP2 lipase using glycerol. Overall, we have developed a set of novel *Yarrowia lipolytica* strong promoters that can be used to engineer strains for improved performance in glycerol-based bioprocesses.

## 2. Materials and Methods

### 2.1. Strains and Culture Conditions

All plasmids (Appendix A) were constructed using commercial *E. coli* competent cells. Mach1-T1 *E. coli* cells (Thermo Fisher Scientific, Waltham, MA, USA) were used for TA cloning, while NEB 5-alpha cells (New England Biolabs, Ipswich, MA, USA) were used for shuttle vector propagation in Luria–Bertani (LB) medium, supplemented with 100 mg/L ampicillin or kanamycin. Agar (15 g/L) was added prior to sterilization for LB agar plates. All bacterial cultures were incubated at 37 °C with shaking at 150 rpm.

All recombinant strains in this study derived from the wild-type *Y. lipolytica* MUCL 28849 (obtained from the Belgian Coordinated Collections of Microorganisms/MUCL Agro-food and Environmental Fungal Collection, Brussels, Belgium) after transformation either with a single replicative vector or a site-specific integration vector. *Y. lipolytica* cells were transformed using the lithium acetate (LiAc) method according to Chen et al. [33]. Successful transformation was verified via PCR amplification of the antibiotic resistance genes *NAT* and *HygR*. Colonies from successfully transformed *Y. lipolytica* were used to inoculate precultures in complex or synthetic media at 30 °C and 150 rpm. The precultures were then used to inoculate 25 mL of corresponding medium in 100 mL shake flasks at an initial OD_600nm_ = 0.25, which served as the assay culture. Cells were grown at 30 °C and 150 rpm shaking. Complex medium contained 10 g/L yeast extract (Oxoid, Hampshire, UK), 20 g/L peptone (Oxoid, Hampshire, UK), and either 20 g/L dextrose (Oxoid, Hampshire, UK) or 20 mL/L glycerol (PENTA Chemicals Unlimited, Prague, Czech Republic), for YPD and YPG media, respectively. The composition of the synthetic medium was described by Josh et al. [34] and was slightly modified [17] (Appendix A).

Yeast media were supplemented, when necessary, with 250 mg/L nourseothricin or 100 mg/L hygromycin. Agar plates were prepared with the addition of 20 g/L agar to YPD or YPG media before sterilization. All agar plates were adjusted with NaOH at pH 7.0.

### 2.2. General Cloning

All restriction enzymes used for cloning purposes were obtained from New England Biolabs (Ipswich, MA, USA), unless specified otherwise. Xpert HighFidelity DNA Polymerase and Xpert Fast DNA polymerase (GRiSP, Porto, Portugal) were used for the PCR amplifications and control reactions. T4 ligation reactions were performed using T4 ligase (New England Biolabs, Ipswich, MA, USA) and incubated at 22 °C for 2 h. Gibson assembly reactions [35] were performed using the Gibson assembly cloning kit (New England Biolabs, Ipswich, MA, USA) according to the manufacturer’s instructions. The reactions were used for transformation into NEB 5-alpha competent *E. coli* cells, according to the recommended heat shock protocol. TA cloning was performed using the TOPO-TA cloning kit (Thermo Fisher Scientific, Waltham, MA, USA), after the addition of A-overhangs to the PCR product, and the reaction was used to transform Mach1-T1 competent *E. coli* cells. Enzymatic reaction purifications and gel extractions were performed using QIAquick PCR Purification Kit and QIAquick Gel Extraction Kit (QIAGEN, Venlo, The Netherlands). Plasmid DNA was extracted using NucleoSpin Plasmid kit (MACHEREY-NAGEL, Düren, Germany). Restriction digestions or PCR amplifications and DNA sequencing were routinely used to verify plasmid constructs.

### 2.3. Construction of Yarrowia lipolytica Expression Vectors

All primers used are listed in the Appendix A. A backbone plasmid capable of autonomous replication in *Y. lipolytica* was created, on which all episomal expression vectors were based. Backbone plasmids pCfB4787, pCfB6576, and pCfB4785 (Addgene plasmids #106147, #106134, and #106149) were kindly provided by Irina Borodina (Technical University of Denmark, Kongens Lyngby, Denmark) [23]. ARS18 DNA sequence [36] was amplified from *Y. lipolytica* genomic DNA using the primers 5′-ARS18/3′-ARS18 (Appendix A) and TOPO-TA-cloned. ARS18 was amplified from pCRII-TOPO/YLARS18 with the primers GBN-ARS18-F/GBN-ARS18-R and cloned by Gibson assembly into pBlueScript SK(-), resulting in pBlue-ARS18. The antibiotic resistance cassettes prTEF1in-NAT-Tcyc1 and prEXP1-HygR-Ttef were amplified from pCfB4787 and pCfB6576 plasmids using the primer pairs GBN-pH-TEF1in-F/GBN-Ts2-R, and GBN-PrEXP1-F/GBN-TTEF-R, respectively, and cloned by Gibson assembly into pBlue-ARS18.

The TEF1in promoter was amplified from pCfB4787 plasmid with primers GBN-pCFB-TEF1in-F/GBN-pCFB-TEF1in-R and cloned through Gibson assembly into pCfB5219. In order to attach the terminator Ts2 [37] (*S. cerevisiae*) to the promoters, TEF1in and EXP1 were subcloned by Gibson assembly into pH-Ts2 (*S. cerevisiae* vector), using the primer pairs GBN-pH-TEF1in-F/GBN-Ts2-R and GBN-pH-EXP1-SmaI-F/GBN-mcs-EXP, respectively. The EXP1 promoter was amplified from pCfB6576. Using the primers GBN-TEF1in-F/GBN-Ts2-R-YL, we were able to amplify the cassette TEF1in-Ts2 for Gibson assembly into pBlueARS18-prEXP1-HygR, resulting in the final pHYLTEF1in episomal vector. The EXP1-Ts2 cassette was amplified with the primers GBN-ARS-EXP1-SmaI/GBN-Ts2-Tef1-R, and Gibson-assembled into pblue-ARS18-PTEF1in-Nat. The resulting expression vector, pNYLEXP1, was used as a backbone to subclone all other promoters by replacing the EXP1 promoter with the 1kb promoter regions, located upstream of each selected gene. All newly identified *Y. lipolytica* promoters along with FBA1in were amplified from MUCL 28849 genomic DNA (gDNA).

FBA1in, consisting of the FBA1 promoter (−848 to −1) along with the 5′ coding region containing the first exon (+1 to +63), the intermediate 102 bp region of the endogenous intron, and the following exon 2 sequence (+166 to +190), was amplified with the primers FBA1p-F/FBA1-E2-R and TOPO-TA-cloned. FBA1in was then cloned into pNYL with Gibson assembly using the primers GBN-ARS18-FBA1in-F/GBN-pNYL-FBA1-E2-R.

Histone 3 (H3) promoter was amplified with the primers YL_H3prom-F/YL_H3prom-R and TOPO-TA-cloned. H3 was amplified from the resulting pCRII-TOPO vector with the primers GBN-pNYL-H3p-F/GBN-pNYL-H3p-R and was Gibson-assembled into pNYL.

ACBP and TMAL promoters were initially amplified using the primer pairs YL-ACBPprom-F/YL-ACBPprom-R and YALI0C06237prom-F/YALI0C06237prom-R and cloned in pCRII-TOPO vectors. Through PCR amplification using the primer YL-ACBPprom-F and the mismatch containing the primer YL-ACBP-X-R, we removed the 3′ XhoI site from the ACBP promoter. Using the primer pair TMAL-trim-XmaI/YTMAL-BamHI-R, we were able to simultaneously excise the 5′ XhoI site and shorten the TMA-Like (TMAL) promoter at the 5′ end. The new ACBP promoter was subcloned into pNYL vector with Gibson assembly using the primers GBN-pNYL-ACBPp-F/GBN-pNYL-ACBPp-R, while the new 837 bp TMALtrim promoter was subcloned into pNYL using BamHI/XmaI restriction sites.

For the subcloning of *mCherry* reporter, the constructed vectors and pCRII-TOPO/mCherry were digested with BamHI/XhoI (or EcoRI/XhoI for subcloning into pNYLACBPp) and *mCherry* was ligated downstream of the promoters.

### 2.4. Construction of Hybrid Promoters

UAS1B8-TEF1(136) was amplified from pCRISPRyl using the primer pair XmaI-UAS1-F/TEF1-BglII-MfeI-XhoI and TOPO-TA-cloned. UAS1B8-TEF1(136) was ligated into pBlueScript SK(-) with XmaI/XhoI restriction sites. Minimal promoters H3(260) and TMAL(250) were amplified from pCRII-TOPO/YLH3p and pCRII-TOPO/TMALtrim with the primer pairs H3p-260-HindIII-F/H3p-260-BglII-MfeI-R and TMALp-250-HindIII-F/TMALp-250-BglII-MfeI-R, respectively, and TOPO-TA-cloned. Using the restriction sites HindIII and MfeI, we removed the TEF1(136) core element and subcloned the H3(260) and TMAL(250) minimal promoters downstream of UAS1B8 sequence. The three hybrid promoters, UAS1B8-TEF1(136), UAS1B8-H3(260), and UAS1B8-TMAL(250), were subcloned into pNYL using XmaI/BglII restriction sites.

### 2.5. Construction of Integration Vectors

All F_3 integrative vectors were constructed through Gibson assembly of the promoters in the backbone pCfB4785. pH3 was amplified from pCRII-TOPO/H3 and pEXP1 was amplified from pCfB6576, using the primer pairs GBN-pCfB-H3-F/GBN-pCfB-H3-R and GBN-pCfB-EXP1-F/GBN-pCfB-EXP1-R, respectively. The primers GBN-pCfB-TMALtr-F/GBN-pCfB-TMALtr-R were used for the amplification of TMALtrim from pNYLTMALtrim. Each hybrid promoter was amplified from the corresponding pBlue-promoter construct using primers GBN-pCfB-Xma-UAS/GBN-pCfB-mcs-R. *mCherry* was subcloned downstream of UAS1B8-TMAL(250), UAS1B8-TEF(136), UAS1B8-H3(260), H3, and EXP1 promoters, with BamHI/XhoI restriction sites. BamHI/EcoRI were used for the subcloning of *mCherry* downstream of the TMALtrim promoter. In addition to the F_3 integrative vectors, pEXP1 was amplified from pCfB6576 using the primers GBN-pCfB-EXP1-F/GBN-pCfB-EXP1-R, and Gibson-assembled into pCfB4787, resulting in the pYLEXP1 INTE_4 integration vector.

The *LIP2* gene was amplified from *Y. lipolytica* MUCL28849 gDNA using the primers YlLIP2-BamHI/YLIP2-R-XhoI, and TOPO-TA-cloned. LIP2 was then subcloned downstream of H3 and UAS1B8-TMAL(250) promoters (in F_3 integrative vectors) using the restriction sites BamHI/XhoI, and downstream of the EXP1 promoter (in the E_4 integrative vector) using the restriction sites BamHI/SalI.

### 2.6. Reverse Transcription-Quantitative PCR

Reverse transcription-quantitative PCR (RT-qPCR) was used to identify highly expressed genes when cells were cultivated in glycerol-containing media. *Y. lipolytica* MUCL 28849 was incubated for 16 h at 30 °C in five different media, i.e., one containing glucose (YPD) and four containing different concentrations of glycerol (YPG with 2% *v*/*v*; YPG with 6% *v*/*v*; complete minimal with 6% *v*/*v*; synthetic medium with 10% *v*/*v*).

RNA was isolated using TRIzol reagent (Thermo Fisher Scientific, Waltham, MA, USA), purified with the RNA Clean & Concentrator^TM^-5 (Zymo Research, Irvine, CA, USA), and finally quantified using the Qubit RNA BR analysis kit (Thermo Fisher Scientific, Waltham, MA, USA). cDNA molecules were prepared by SuperScript II Reverse Transcriptase (Thermo Fisher Scientific, Waltham, MA, USA) according to the manufacturer’s recommendations. The qPCR reactions were conducted using Luna Universal qPCR Master Mix reagent (New England Biolabs, Ipswich, MA, USA) in StepOnePlus Real-Time PCR System (Thermo Fisher Scientific, Waltham, MA, USA) to monitor the amplification of specific DNA sequences during the reaction. Thus, the expression levels of the genes *H3* (histone 3), *ACBP* (acyl-CoA-binding protein), *AQP* (aquaporine-like protein), *DIOX* (dioxygenase), *GK* (Glycerol kinase), and *TMAL-like* (trimethylamine-like) were examined using specific primers for those target genes (Appendix A). The constitutively expressed *TEF1* native gene was used as reference gene [38].

### 2.7. Bioinformatic Analysis of Public Transcriptome Data

Raw data (SRR9021304, SRR9021305, SRR9021306, SRR9021307, SRR9021308, and SRR9021313—BioProject PRJNA437435) were downloaded from the SRA-NCBI and were trimmed and quality-filtered using TrimGalore (v0.6.7) [39]. After trimming, adapter sequences, ambiguous bases (N), and reads with quality scores below 30 were removed from the data. Differential expression analysis was performed according to the “new Tuxedo” pipeline [40]. Specifically, filtered reads were aligned against the *Y. lipolytica* reference genome (GCF_000002525.2) using HISAT2 (v2.1.0). Transcript assembly and quantification was performed with StringTie (v2.0.6). Output data were imported into R (v4.0.1) [41] and differential expression analysis was conducted using the Ballgown package (v2.28.0). Differentially expressed genes (DEGs) were considered those with *p*-value < 0.005 and |log2(FC)| > 2. The visualization of the results was performed using the ggplot2 (v3.3.6) [42] and pheatmap (v1.0.12) [43] packages.

### 2.8. Flow Cytometry and Fluorescence Microscopy

Yeast cultures were incubated for 96 h and 300 μL samples were collected at 24 h intervals. Prior to flow cytometry analysis, 20 uL of each cell culture was diluted in 980 uL of Phosphate-Buffered Saline (PBS), pH 7.4. Flow cytometry analysis was performed in BriCyte E6 flow cytometer (Mindray Bio-Medical Electronics, Nanshan, China), and the data were analyzed using FlowJo™ v10 (BD Life sciences, Franklin Lakes, NJ, USA). Appropriate gates were applied in order to select single cells (Appendix A). The geometric mean of the generated data was used to calculate the fluorescence intensity of each sample. The number of events recorded was set to 50.000. For the fluorescence microscopy, cells were centrifuged at 5000× *g* for 2 min, washed with dH_2_O, and resuspended in PBS before being imaged with the ZOE Fluorescent Cell Imager (Bio-Rad Laboratories, Hercules, CA, USA).

### 2.9. Determination of Lipase Activity

The activity of the LIP2 lipase was assayed using a colorimetric method based on monitoring the released *p*-nitrophenol (pNP) using *p*-nitrophenyl butyrate (pNPB) as substrate [44]. An amount of 1 mL of a 48 h main culture, either in YPG or SM medium (stationary phase), prepared as described in 2.1, was centrifuged for five minutes at 18.400× *g*, and the supernatant, containing the secreted LIP2, was transferred to a 96-well microplate and used for the hydrolysis of pNPB (5 mM in DMSO). Whenever needed, the supernantant was diluted using the appropriate amount of sodium phosphate buffer (100 mM). Lipase activity was measured spectrophotometrically at 410 nm (TECAN SPARK^®^ multimode microplate reader, Tecan Trading AG, Männedorf, Switzerland) by following the hydrolysis of pNPB at 30 °C for 460 s. One unit of lipase activity was defined as the amount of enzyme releasing 1 µmol pNP per minute using pNPB as a substrate at 30 °C and pH 7.6.

## 3. Results

### 3.1. Identification of Highly Expressed Genes in Y. lipolytica Cultured in Glycerol Media

In order to identify new promoter elements that maintain high expression in media containing either pure or biodiesel-derived crude glycerol, available public raw data from the SRA-NCBI (SRR9021304, SRR9021305, SRR9021306, SRR9021307, SRR9021308, and SRR9021313—BioProject PRJNA437435) from cells growing in YPD and YPG were employed. After trimming, adapter sequences, ambiguous bases (N), and reads with quality scores below 30 were removed from the data. Differential expression analysis was performed by aligning the filtered reads against the *Y. lipolytica* reference genome (GCF_000002525.2). Transcript assembly and quantification were performed with StringTie (v2.0.6). Output data were imported into R [41] and differential expression analysis was conducted. Differentially expressed genes (DEGs) were considered those with *p*-value < 0.005 and |log2(FC)| > 2. Visualization of the results clustered the generated data into two distinct groups based on the carbon source of the medium, i.e., glycerol or glucose (Figure 1A).

Based on the in silico differential expression analysis, six genes exhibiting very high expression in glycerol-based media were selected for a further comparison of their transcript levels: histone 3 (*H3*, Gene ID: 2907691), acyl-CoA binding protein (*ACBP*, Gene ID: 7009567), trimethylamine-like (*TMAL*, Gene ID: 2909241), dioxygenase *(DIOX*, Gene ID: 2906426), glycerol kinase (*GK*, Gene ID: 2908247), and aquaporin-like protein (*AQP*, Gene ID: 2908215). Triplicate cultures of *Y. lipolytica* wild-type MUCL 28849 cells were incubated for 16 h at 30 °C, in glucose-based (YPD with 2% *w*/*v*) and glycerol-based media (YPG with 2% *v*/*v*; YPG with 6% *v*/*v*; complete medium (CM) with 6% *v*/*v*; synthetic medium (SM) with 10% *v*/*v*). RNA was extracted and RT-qPCR was performed to assess gene expression levels. Among the genes tested, three of them showed high expression levels (*H3*, *ACBP*, and *TMAL*). The expression level of the *H3* gene was three-fold higher than the second best, *ACBP*. Expression of *H3* and *ACBP* was significantly elevated in YPG with 2% *v*/*v* glycerol when compared to YPD, coinciding with the transcriptome data. At higher glycerol concentrations, gene expression noticeably decreases, but still remains at high levels. The expression levels of all tested genes were similar in the SM with 10% *v*/*v* glycerol and YPG with 6% *v*/*v* glycerol, rendering the former a potent inexpensive equivalent of YPG medium (Figure 1B).

### 3.2. Development of Novel Promoter–Reporter Constructs

The H3, ACBP, and TMAL promoter elements were cloned by amplifying the 1 kb region upstream of the ATG site. Episomal and integrative vectors were developed carrying the H3, ACBP, and TMAL promoters and the previously characterized strong promoters EXP1, TEF1in, and FBA1in for benchmarking. To generate the episomal plasmid vectors, we integrated into a bacterial plasmid backbone the *Y. lipolytica* autonomously replicating sequence element ARS18 [39], the antibiotic resistance marker cassettes pTEF1in-NAT-Tcyc1 or pEXP1-HygR-Ttef, a multicloning site, and the synthetic terminator Tsynth2 (Ts2) [45] by GIBSON assembly (Figure 2A). The promoter elements H3, ACBP, TMAL, TEF1in, FBA1in, and EXP1 were cloned upstream of the multiple cloning site (MCS). The reporter fluorescent *mCherry* gene was inserted into the MCS to generate the final episomal constructs. The *Y. lipolytica* episomal plasmids offer the advantage of locus-independent expression but they are maintained in only one to three copies per cell and do not have the stability of *S. cerevisiae* plasmids. For this reason, most recombinant genes are expressed from integrative cassettes. To develop the integrative promoter constructs, the EasyCloneYALI-Integrative Vector Set (Addgene Kit #1000000140) was utilized [23]. The pCfB4785 plasmid contains flanking integration sequences F_3 (chromosome F) and two termination sequences *pex20* and *lip2*, enabling the cloning of dual promoters, and the cassette pTEF1in-NAT-Tcyc1 flanked by LoxP sites, which allow the excision of the selection marker with Cre recombinase. All the promoter elements above were inserted together with cloning sites into the pCfB4785 plasmid. The *mCherry* gene was subsequently subcloned into each plasmid (Figure 2B).

### 3.3. Reporter Gene Expression Driven by Selected Promoters

The generated episomal constructs carrying the selected endogenous promoters were transformed into *Y. lipolytica* MUCL 28849 cells and selected using the appropriate antibiotic (nourseothricin or hygromycin). Transformants were grown in flasks with 25 mL YPG medium supplemented with antibiotic and mCherry fluorescence was measured at 24, 48, 72, and 96 h to assess promoter strength. Autofluorescence was measured at the same time points in untransformed (wild-type) cells. The results show that all three selected novel endogenous promoters were very strong (pH3 > pEXP1 > pTMAL > pACBP > pFBA1in > pTEFin) compared to the benchmarking promoters pFBA1in and pTEFin. The last two promoters exhibited very weak promoter activity in our constructs in the YPG medium (Figure 3A). These results were corroborated in the widely used *Y. lipolytica* PoId (CLIB139) strain (Appendix A). Interestingly, pH3 and pEXP1 seem to be very strong at an early time point (24 h) compared to pTMAL, which expresses at very high levels at 72 and 96 h. For the integrative cassette expression measurements, we selected only the three strongest promoters pH3, pEXP1, and pTMAL, expecting fluorescence to be much weaker in strength due to their single-copy integration (Figure 3B). The data confirmed the observations from episomal constructs showing that pH3 is better than pEXP1, which is equivalent to pTMAL. The early onset of fluorescence of pH3 at 24 h is also observed in this case. Fluorescence from the episomal constructs can be easily observed in fluorescent microscopy, where merged images reveal two distinct groups of fluorescent and non-fluorescent cells (Figure 3C).

### 3.4. Development of Hybrid Promoters

Aiming to further increase promoter strength, we also pursued the development of hybrid promoters based on our newly isolated promoter elements. The previously characterized UAS1B element from the XPR2 promoter fused in tandem eight times was ligated to the 260 bp core promoter from pH3 and the 250 bp from pTMAL. The constructed hybrid promoters were inserted into episomal and integrative vectors and benchmarked against the UAS1B8-TEF(136) hybrid promoter, currently used by various labs for high expression, and the full endogenous H3 and TMAL promoters. The two new hybrid promoters, UAS1B8-H3(260) and UAS1B8-TMAL(250), were substantially stronger than all other promoters and double in strength compared to UAS1B8-TEF(136) (Figure 4A). This is also quite evident in the integrated expression cassettes, where the novel hybrid promoters differentiate significantly from background levels (Figure 4B). Promoter activity is confirmed by fluorescence microscopy where the number and intensity of fluorescent cells are much higher than the UAS1B8-TEF(136) expression construct (Figure 4C).

### 3.5. Effect of Carbon Source on the Promoter Strength

We subsequently tested promoter expression in glucose-based YPD medium and compared it to the corresponding glycerol YPG medium. The fluorescence of episomal pACBP, pTMAL, and pH3 reporter constructs was at similar levels in both media, though at 24 h, pTMAL and pH3 in YPD is slightly higher. The pEXP1 construct showed a modest increase in YPD medium for all time points, while FBA1in remained at the same low levels (Figure 5A). For the hybrid promoters, UAS1B8-H3(260) exhibited a 20–40% increase in YPD at later time points, UAS1B8-TMAL(250) a 40–60% increase in YPD, whereas the UAS1B8-TEF(136) remained the same (Figure 5C). When the integrated reporter constructs were compared for fluorescence in YPD and YPG media, pTMAL INTF_3 and pH3 INTF_3 were elevated in YPD at 48 and 72 h but leveled off at 96 h. pEXP1 INTF_3 expressed slightly better in YPD (Figure 5B). In the case of the two integrated hybrid promoters, though, a substantial increase in fluorescence was observed in YPD, which reached the maximum at 96 h. UAS1B8-H3(260) INTF_3 is two-fold higher at 72 h and three-fold at 96 h. UAS1B8-TMAL(250) INTF_3 is 1.75-fold higher at 72 h and 2.5-fold at 96 h in YPD media. A comparatively lower increase was observed for UAS1B8-TEF(136) INΤF_3, reaching 1.7-fold at 72 h and 2-fold at 96 h (Figure 5D).

### 3.6. Effect of Synthetic Medium on Promoter Strength

Finally, promoter activity after the genomic integration of the expression cassetes was tested in glycerol-based synthetic medium (SM), which would be a desirable inexpensive medium for industrial applications. The expression of *mCherry* was dramatically decreased in SM compared to other tested complex media. Only the pH3, pEXP1, and UAS1B8-H3(260) showed any minimal fluorescence above background levels at 24 and 48 h, diminishing at later time points (Figure 6).

### 3.7. Expression of LIP2 from Integrated Expression Constructs

To test the suitability of the newly constructed integrated expression cassettes for biotechnological applications, the native *LIP2* gene was selected, encoding for the extracellular lipase LIP2. For that, recombinant *Yarrowia* strains were generated, bearing (i) pEXP1-LIP2 INTE_4 (one gene copy); (ii) pH3-LIP2 INTF_3 (one gene copy); (iii) pEXP1-LIP2 INTE_4 and pH3-LIP2 INTF_3 (two gene copies); and (iv) UAS1B8-TMAL(250) INTF_3 (hybrid promoter, one gene copy). As seen in Figure 7A,B, in all cases, neither the type of promoter nor the number of *LIP2* gene copies had a negative impact on the growth rate of the recombinant strains. In the case of YPG, the growth rate of all strains ranged between 0.43 and 0.44 h^−1^; however, the strain bearing the hybrid promoter yielded the lowest biomass of all, hinting towards a heavier metabolic burden for the strain under this condition (Figure 7A). In SM supplemented with 7.5% *v*/*v* glycerol, all strains grew with a rate of 0.29–0.31 h^−1^, significantly lower compared to that in complex medium. It is clear that when SM is used, the final biomass of all strains is also dramatically decreased, being almost half of that reached in the complex medium (Figure 7B). It seems that even though the YPG complex medium contains a lower glycerol concentration (2% *v*/*v*), the presence of an organic nitrogen source is more advantageous than the inorganic one of the SM [46]. The increased biomass formation of all strains when using complex instead of synthetic media increased the hydrolytic activities of the corresponding LIP2-containing supernatants (Figure 7C,D). It should be noted that the usage of a synthetic medium might have a negative impact not only on the biomass formation but also on the expression levels of *LIP2* itself, as also shown in the flow cytometry results obtained with the reporter protein mCherry (Figure 6). The superiority of the hybrid promoter with one *LIP2* gene copy is apparent in both complex and synthetic media, even compared with the strain bearing two copies of the *LIP2* gene, leading to 3.4- and 1.4-fold higher hydrolytic activity in Units/mL_supernatant_, respectively, with one unit (U) defined as the activity releasing 1 µmol pNP per minute at 30 °C (Appendix A).

## 4. Discussion

Over the last two decades, numerous genetic tools for *Yarrowia lipolytica* have been developed and applied in the engineering of the microorganism. However, limitations still exist compared to the tools available for *S. cerevisiae*. One such limitation is the scarcity of strong promoters that can be functional in glycerol and synthetic media, hindering the full exploitation of the yeast’s capacity to catabolize various inexpensive carbon sources. This need becomes pronounced due to the lack of stable episomal high-copy plasmids. Using public transcriptome data from cells growing in YPG and YPD, we identified three strong native promoters (pH3, pACBP, and pTMAL) with excellent expression in glycerol media. The cloned promoters matched or exceed the expression levels of the strongest EXP1 native promoter in YPD, YPG, and SM, enriching the palette of strong promoters. As a response to the need for strong promoters, several groups developed hybrid promoters composed of a core element (TATA box and ATG region) or a minimal promoter element (usually 130–260 bp upstream of ATG) and Upstream Activator Sequences (UAS) [25,26,31]. Several UASs have been identified and characterized in *Yarrowia lipolytica*. Two such frequently used UASs originated from the POX2 gene, which encodes an acyl-CoA oxidase involved in fatty acid metabolism, and the nitrogen- and pH-regulated XPR2 promoter, which gave rise to a frequently used set of hybrid promoters that contain the UAS1B element. When used in tandem, the UAS1B elements provide an enhancement in expression that is independent of nitrogen level and pH [25,26,31,47]. The hybrid UAS1B8-TEF(136) and UAS1B8-TEF(203) promoters are considered very strong, achieving much higher transcription levels than the native promoter elements [26]. In our hands, when integrated into the genome, the UAS1B8-TEF(136) promoter is also stronger than the native promoters tested. When we exchanged the TEF minimal promoter with the TMAL(250) and H3(260) promoter fragments, which are dense in TF binding sites as identified using the YEASTRACT online tool [48], transcriptional activity rose to dramatically higher levels. This observation was also confirmed upon the expression of *LIP2* from the hybrid UAS1B8-TMAL(250), where lipolytic activity is significantly higher even compared to a double integration construct containing the strongest endogenous promoters, EXP1 and H3.

However, despite the high transcriptional activity of the hybrid promoters, when SM was used, the expression levels were equivalent or lower compared to the native promoters. Future work could address modifications to the composition of SM that induce gene expression, or alternatively search for a UAS that could be responsive even in poor synthetic media. Additionally, the H3(260) minimal promoter could be utilized to generate a dual hybrid promoter that would express equivalently high levels of the transgenes. The set of integration vectors developed by Holkenbrink et al. [23], which contains two terminators, would be ideal.

## 5. Conclusions

Novel strong endogenous *Y. lipolytica* promoters (ACBP, H3, TMAL), which drive high expression of transgenes in glycerol- and glucose-based media, were identified and characterized. Hybrid promoters developed from the TMAL(250) and H3(260) minimal promoters express two- to three-fold higher levels than the benchmark UAS1B8-TEF(136) promoter and can be used to superinduce expression of target genes in *Y. lipolytica* engineering projects.

## Figures and Tables

**Figure 1 microorganisms-11-01152-f001:**
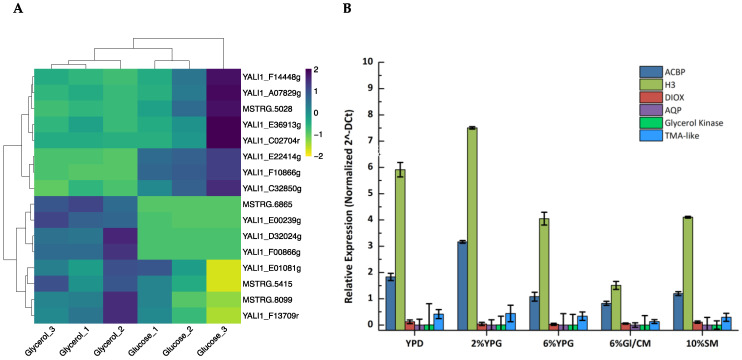
Effect of the carbon source on highly expressed genes in *Yarrowia lipolytica.* (**A**) Heatmap of differencially expressed genes when *Yarrowia lipolytica* cells grow on glycerol or glucose. (**B**) Relative expression of the six selected highly expressed genes in one glucose- (YPD 2% *w*/*v*) and four glycerol-based growth media (YPG 2% *v*/*v*; YPG 6% *v*/*v*; CM, complete medium 6% *v*/*v*; SM, synthetic medium 10% *v*/*v*). The data represent mean ± SD of three independent biological replicates.

**Figure 2 microorganisms-11-01152-f002:**
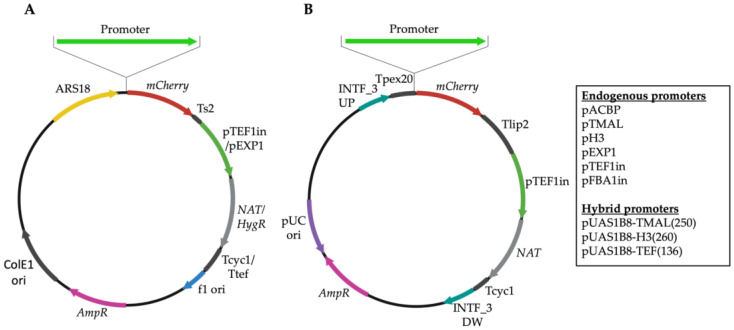
Maps of the plasmids used in this study: (**A**) episomal plasmids were constructed by assembling the antibiotic resistance expression cassettes NAT/HygR, the autonomously replicating sequence ARS18, the endogenous promoters and hybrid promoters, a reporter mCherry cDNA, and a synthetic termination sequence Ts2; (**B**) integrative cassettes were constructed based on the pCfB4785 plasmid which contains flanking integration sequences F_3 (chromosome F) and the various promoters with the *mCherry* reporter gene.

**Figure 3 microorganisms-11-01152-f003:**
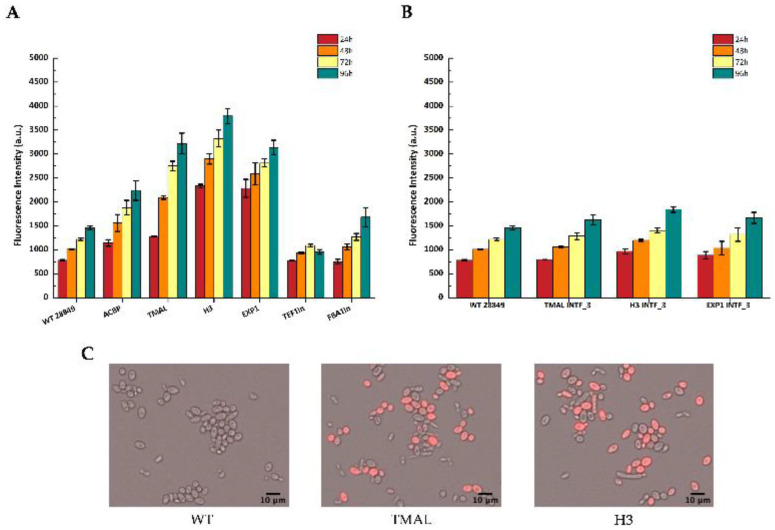
Activity of endogenous promoters as determined by flow cytometry and presented as fluorescence intensity of the reporter mCherry protein after cultivation in YPG medium for 96 h. WT 28849 corresponds to untransformed *Y. lipolytica* MUCL28849 cells. The data represent mean ± SD of three independent biological replicates. (**A**) Activity of the endogenous promoters in episomal vectors; (**B**) activity of the strongest endogenous promoters H3, TMAL, and EXP1 after genomic integration (INTF_3); (**C**) fluorescence microscopy of *Y. lipolytica* strains from 72 h cultures, with episomal vectors carrying different endogenous promoters: WT (left), pTMAL (middle), and pH3 (right). The images were created by merging the respective bright field and red channel images.

**Figure 4 microorganisms-11-01152-f004:**
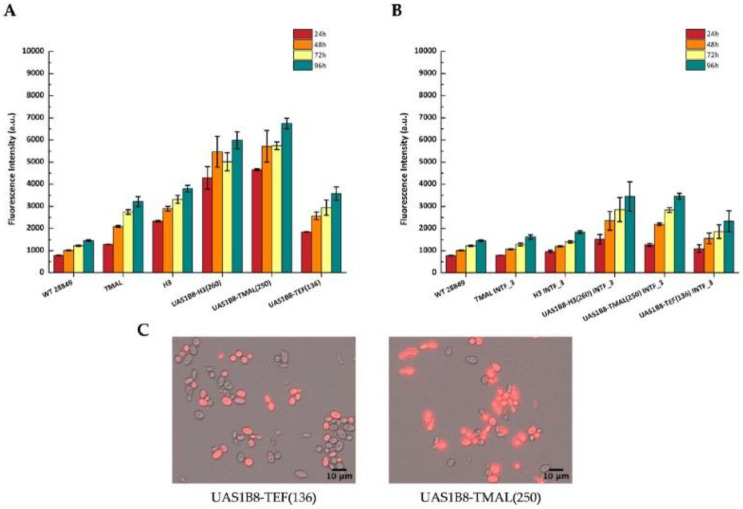
Activity of hybrid promoters as determined by flow cytometry and presented as fluorescence intensity of the reporter mCherry protein after cultivation in YPG medium for 96 h. WT 28849 corresponds to untransformed (wild-type) *Y. lipolytica* MUCL28849 cells. The data represent mean ± SD of three independent biological replicates. (**A**) Activity of hybrid promoters in episomal vectors; (**B**) activity of the hybrid promoters after genomic integration (INTF_3), data from pTMAL and pH3 were used for comparison; (**C**) fluorescence microscopy of *Y. lipolytica* strains from 72 h cultures, with episomal vectors carrying different hybrid promoters: UAS1B8-TEF(136) (left) and UAS1B8-TMAL(250) (right). The images were created by merging the respective bright field and red channel images.

**Figure 5 microorganisms-11-01152-f005:**
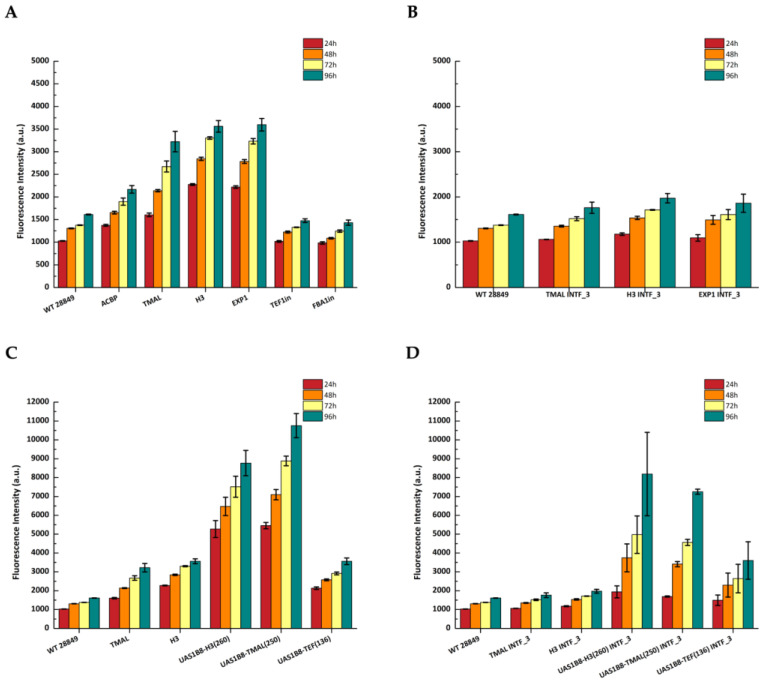
Promoter activity as determined by flow cytometry and presented as fluorescence intensity of the reporter mCherry protein after cultivation in YPD medium for 96 h. (**A**) Activity of the endogenous promoters in episomal vectors; (**B**) activity of the strongest endogenous promoters H3, TMAL, and EXP1 after genomic integration (INTF_3); (**C**) activity of the hybrid promoters in episomal vectors; (**D**) activity of the hybrid promoters after genomic integration (INTF_3). WT 28849 corresponds to untransformed (wild-type) *Y. lipolytica* MUCL28849 cells. The data represent mean ± SD of three independent biological replicates.

**Figure 6 microorganisms-11-01152-f006:**
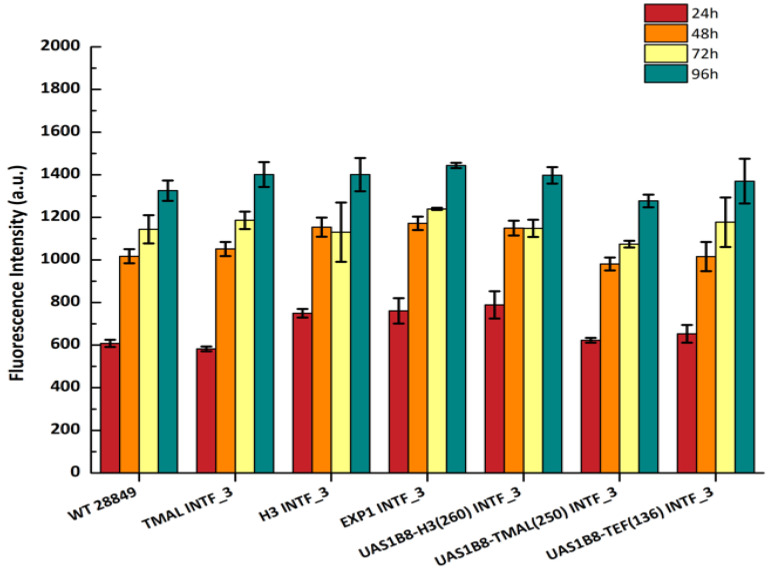
Activity of all integrated promoters in SM. Promoter activity was determined by flow cytometry and presented as fluorescence intensity of the reporer mCherry protein after cultivation for 96 h. WT 28849 corresponds to untransformed (wild-type) *Y. lipolytica* MUCL28849 cells. The data represent mean ± SD of three independent biological replicates.

**Figure 7 microorganisms-11-01152-f007:**
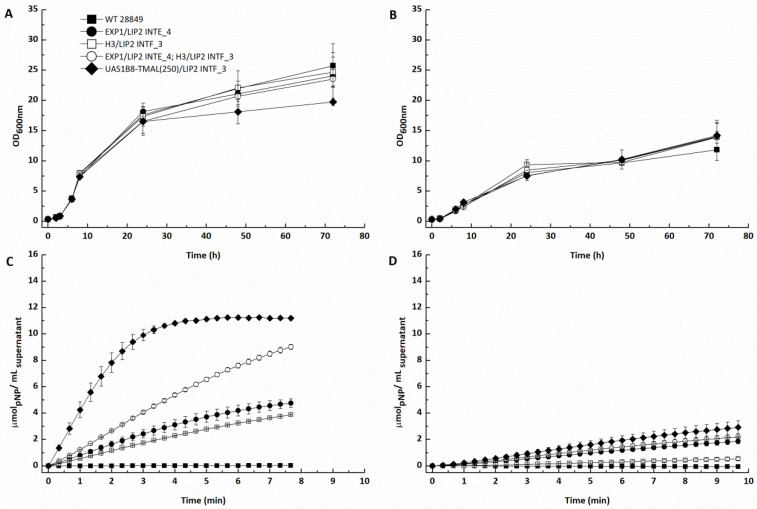
Growth profile (OD_600nm_) of recombinant *Yarrowia* strains bearing the newly constructed integrated *LIP2* expression cassetes: (**A**) effect of promoter type and *LIP2* copy number on growth using YPG medium at 30 °C and 150 rpm; (**B**) effect of promoter type and *LIP2* copy number on growth using SM medium with 7.5% *v*/*v* glycerol at 30 °C and 150 rpm. Monitoring the hydrolytic activity (μmol_pNP_/mL_supernatant_) of (**C**) YPG and (**D**) SM culture supernatants containing the secreted LIP2 by the recombinant *Y. lipolytica* strains. Hydrolytic activity was photometrically (410 nm) determined using the pNPB assay as described in Materials and Methods and expressed as the μmol pNP released at 30 °C by LIP2 secreted in 1 mL culture medium. WT 28849 corresponds to untransformed (wild-type) *Y. lipolytica* MUCL28849 cells.

## Data Availability

All data are included in the manuscript and Appendix A.

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
