# Peer review of "Identification and Construction of Strong Promoters in Yarrowia lipolytica Suitable for Glycerol-Based Bioprocesses"

_microorganisms, 2023, doi:10.3390/microorganisms11051152_

Round 1
Reviewer 1 Report
The scientific article with title: “Identification and construction of strong promoters in Yarrowia lipolytica suitable for glycerol-based bioprocesses”, submitted by the authors: Ioannis Georgiadis, Christina Tsiligkaki, Victoria Patavou, Maria Orfanidou, Antiopi Tsoureki, Aggeliki Andreadelli, Eleni Theodosiou, and Antonios M. Makris as corresponding author, describes the identification and characterization of Yarrowia lipolytica strong native and synthetic promoters. The reported promoters are shown to be able to drive strong gene expression in different culture media, and subsequently, they could be used for efficient metabolic engineering of this widely used microorganism.
The topic of the article is very relevant to this journal as it describes new strategies to boost metabolic pathways and heterologous expression from Yarrowia. The use of novel strong promoters that drive efficiently genes encoding heterologous biosynthetic enzymes, can facilitate and expand the use of this unique host and potential production platform in different culture media. The work is nicely presented, and the results and conclusions are very well supported by the reported experiments.
Therefore, I have only minor comments for the present manuscript:
1. In figure 1A, I believe it would be useful to show the genes/transcripts of the Y axis, so readers can relate them to the selected highly expressed genes (shown in Fig 1B).
2. Figure 3: the exact nature of WT28849 is not described. Is it the mCherry with no promoter? Is it an empty vector? I believe the authors should describe this. Same for WT in Fig. 3C.
3. Figure 4C and lines 411-413: in the cells transformed with the UAS1B8-TEF(136) we observe more cells with no fluorescence, when compare with the UAS1B8-TMAL(250). This does not necessarily means that the UAS1B8-TMAL(250) promoter is stronger, but rather that the episomal construct expressing mCherry with this promoter is more stable (?), or the transformation efficiency was higher. I believe the authors should explain their findings more in detail, and they should mention how they interpret these results.
4. Lines 525-527: “The hybrid UAS1B8-TEF(136) and TEF(203) promoters are considered very strong promoters achieving much higher transcription levels than the native promoter elements”. Reading the article, I got the impression that TEF is a native promoter. If this stands true, then the sentence should be rephrased.
Author Response
- In figure 1A, I believe it would be useful to show the genes/transcripts of the Y axis, so readers can relate them to the selected highly expressed genes (shown in Fig 1B).
R: The figure was revised accordingly.
- Figure 3: the exact nature of WT28849 is not described. Is it the mCherry with no promoter? Is it an empty vector? I believe the authors should describe this. Same for WT in Fig. 3C.
R: The sentence “WT 28849 corresponds to untransformed Y. lipolytica MUCL28849 cells” has been added to the legend of the figure 3 (L395).
- Figure 4C and lines 411-413: in the cells transformed with the UAS1B8-TEF(136) we observe more cells with no fluorescence, when compare with the UAS1B8-TMAL(250). This does not necessarily means that the UAS1B8-TMAL(250) promoter is stronger, but rather that the episomal construct expressing mCherry with this promoter is more stable (?), or the transformation efficiency was higher. I believe the authors should explain their findings more in detail, and they should mention how they interpret these results.
R: The presence of heterogeneous populations of fluorescing and non-fluorescing cells is also observed in other organisms, such as episomal expression of fluorescent reporters in S. cerevisiae. The most likely explanation is due to natural cell heterogeneity and secondarily to instability issues. Expression from strong promoters increases the intensity of fluorescence and more cells are seen fluorescing, likely because some of the cells that were expressing very small levels of reporter, under a stronger promoter exceed the detectability levels.
- 4. Lines 525-527: “The hybrid UAS1B8-TEF(136) and TEF(203) promoters are considered very strong promoters achieving much higher transcription levels than the native promoter elements”. Reading the article, I got the impression that TEF is a native promoter. If this stands true, then the sentence should be rephrased.
R: The sentence (L533) has been rephrased to clarify that we are referring to 2 hybrid promoters, UAS1B8-TEF(136) and UAS1B8-TEF(203)
Reviewer 2 Report
Reviewer report on manuscript ‘Identification and construction of strong promoters in Yarrowia lipolytica suitable for glycerol-based bioprocesses’
In this work, authors identified strong native promoters in glycerol-based media, and six highly expressed genes were mined from public data, analyzed, and validated. The promoters from the three most highly expressed were cloned upstream of the reporter mCherry in episomal and integrative vectors. Fluorescence was quantified by flow cytometry and promoter strength was benchmarked with known strong promoters in cells growing in glucose, glycerol, and synthetic glycerol media. The results show that they are very strong promoters with pH3 exceeding all other tested promoters. The new hybrid promoters exhibited far superior strength. This research identified and characterized several strong Y. lipolytica promoters that expand the capacity to engineer Yarrowia strains and valorize industrial byproducts.
The manuscript is well supported by supplementary data, it is well written and could be interesting for researchers who are working in the area of microbiology. Therefore, the manuscript can be published after the below-indicated minor corrections and improvements:
The justification of research is partially addressed in the initial part of the introduction, but the justification can be strengthened and advanced towards the importance of application in glycerol oxidation, which is important in biosensors and especially in biofuel cells (From Microorganism-based Amperometric Biosensors towards Microbial Fuel Cells. Sensors 2021, 21, 2442.).
Conclusions are very strong and not very efficient, therefore, conclusions can be advanced and improved.
Minor editing of English language required.
Author Response
The justification of research is partially addressed in the initial part of the introduction, but the justification can be strengthened and advanced towards the importance of application in glycerol oxidation, which is important in biosensors and especially in biofuel cells (From Microorganism-based Amperometric Biosensors towards Microbial Fuel Cells. Sensors 2021, 21, 2442.).
R: Citation has been incorporated in L54.
Conclusions are very strong and not very efficient, therefore, conclusions can be advanced and improved.
R: Changes were made accordingly.